# Learning Pipelines with Limited Data and Domain Knowledge: A Study in Parsing Physics Problems

**Mrinmaya Sachan**♣   **Avinava Dubey**♣   **Tom Mitchell**♣   **Dan Roth**♠   **Eric P. Xing**♣◇

♣Machine Learning Department, School of Computer Science, Carnegie Mellon University
♠Department of Computer and Information Science, University of Pennsylvania
◇Petuum Inc.
{mrinmays,akdubey,tom.mitchell,epxing}@cs.cmu.edu
danroth@seas.upenn.edu

## Abstract

As machine learning becomes more widely used in practice, we need new methods to build complex intelligent systems that integrate learning with existing software, and with domain knowledge encoded as rules. As a case study, we present such a system that learns to parse Newtonian physics problems in textbooks. This system, `Nuts&Bolts`, learns a pipeline process that incorporates existing code, pre-learned machine learning models, and human engineered rules. It jointly trains the entire pipeline to prevent propagation of errors, using a combination of labelled and unlabelled data. Our approach achieves a good performance on the parsing task, outperforming the simple pipeline and its variants. Finally, we also show how `Nuts&Bolts` can be used to achieve improvements on a relation extraction task and on the end task of answering Newtonian physics problems.

## 1   Introduction

End-to-end learning is the new buzz word in machine learning. Models trained in an end-to-end manner have achieved state-of-the-art (SOTA) performance on various tasks like image classification, machine translation and speech recognition. However, a common barrier for using end-to-end learning is the amount of data needed to train the model. For reference, the SOTA image classification model, VGGNet [43], is trained on 1.2M images with category labels and the SOTA machine translation model, GNMT [52], is trained on a dataset of 6M sentence pairs, 340M words. One possible remedy to the issue of data-hungriness is to incorporate domain knowledge. However, due to the very nature of the methods used, incorporating domain knowledge in end-to-end learning is challenging [47].

In contrast, pipelines [51] decompose a complex task into a series of easier-to-handle sub-tasks (stages), where the local predictor at a particular stage depends on predictions from previous stages. Pipelines can be tuned with small amount of labeled data and it is easier to incorporate domain knowledge expressed as rules, existing software and pre-learnt components. However, pipelining suffers from propagation of local errors [11].

Thus, we propose `Nuts&Bolts`: an approach for learning pipelines with labeled data, unlabeled data, existing software and domain knowledge expressed as rules. By jointly learning the pipeline, `Nuts&Bolts` retains the advantages of end-to-end learning (i.e. doesn't suffer from error propagation). Furthermore, it allows for easy incorporation of domain knowledge and reduces the amount of supervision required, removing the two key shortcomings of end-to-end learning.

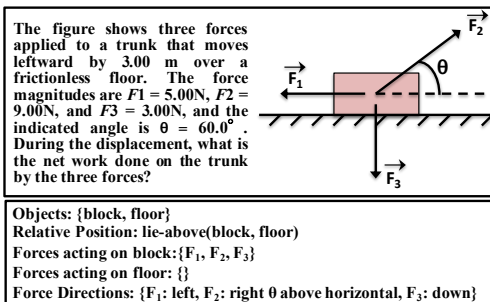

The figure shows three forces applied to a trunk that moves leftward by 3.00 m over a frictionless floor. The force magnitudes are $F1 = 5.00$N, $F2 = 9.00$N, and $F3 = 3.00$N, and the indicated angle is $\theta = 60.0°$. During the displacement, what is the net work done on the trunk by the three forces?

Objects: {block, floor}
Relative Position: lie-above(block, floor)
Forces acting on block:{$F_1$, $F_2$, $F_3$}
Forces acting on floor: {}
Force Directions: {$F_1$: left, $F_2$: right $\theta$ above horizontal, $F_3$: down}

Figure 1: Above: An example Newtonian physics problem. Below: Diagram parsed in formal language.

We are motivated by the novel task of parsing Newtonian physics problems into formal language (see Figure 1). This is useful as it builds a computer ingestable rich semantic representation of

these problems. It is also a key step in a large line of concurrent research on building a solver for such problems [41, 17, 38, 37, 39]. The problems typically consist of a paragraph of text and (often) an associated diagram. These problems are quite diverse, representing complex physical and mathematical concepts which are not often present in natural text and images[1] – understanding them itself requires substantial domain knowledge. Hence, traditional NLP and Computer Vision methods cannot be directly ported to extract and represent the semantics of these problems, and a richer integration of domain knowledge is needed. We model `Nuts&Bolts` using logic programming and show improvements over multiple baselines as well as various pipeline approaches, achieving state-of-the-art results.

Our paper is structured as follows: we first introduce the task of parsing Newtonian physics problems in section 2 and use it to motivate our approach in section 3. Then, we describe the parsing pipeline in section 4, and our experiments in section 5. We review related work in section 6. Then, finally we will showcase the benefit of our approach on the problem of relation extraction from text in section 7 where we achieve state-of-the-art results outperforming *Snorkel*, a strong relation extraction system.

## 2   Motivation

**Problem Definition:** To motivate our work, we first define the task of parsing Newtonian physics problems as one of mapping the problem (text and diagram) to a formal logical language. We choose our formal language as a subset of typed first-order logic comprising of `constants` (3.00 m, 5.00 N, 60°, etc.), `variables` ($F_1$, $\theta$, etc.), and a hand-picked set of 138 predicates (`equals`, `mass`, `distance`, `force`, `speed`, `velocity`, `work`, etc.). Each element in the language is strongly typed. For example, the predicate `mass` takes the first argument of the type "object" and the second argument of the type "mass-quantity" such as `2kg`, `3g`, etc. As shown in Figure 1, the parse is represented as logical formulas, i.e. conjunctions over applications of (possibly negated) predicates with arguments which can be constants or variables.

The primary motivation behind modelling this task as a pipeline is the difficulty of solving such a challenging problem with a single monolithic learner; that expressing this problem directly in terms of input and output will result in a complex function that may be impossible to learn. At the same time, there is a high cost associated with obtaining sufficient labeled data to achieve good learning performance which is difficult in such niche domains. However, the pipeline allows us to think of the task in a modular way, and to integrate stagewise supervision and domain knowledge of physics into the model. It also allows us to supervise the various sub-components to aid rapid learning.

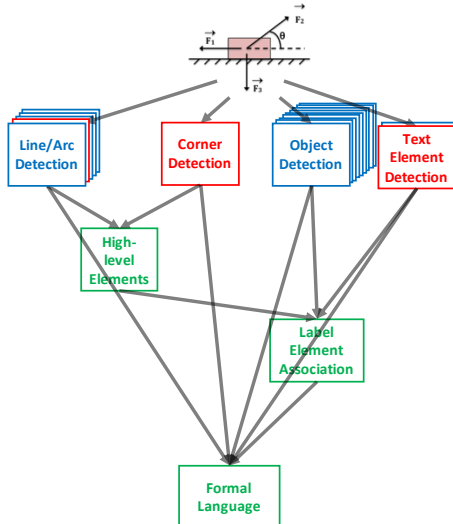

Figure 2: A pipeline for diagram parsing with various (possibly multiple) pre-trained functions, existing software and rules. The pre-learnt components are shown in blue, the existing software are shown in red and rule-based components are shown in green.

We begin by describing the parsing pipeline at a high level. We break the parsing task into three phases. In the first phase, we parse the diagram recognizing the various diagram elements and relationships between them, leading to a diagram parse in formal language. In the second phase, we parse the problem text into the same formal language. In the third and final phase, we reconcile the diagram and the text parse and achieve the final parse of the problem[2]. Diagram parsing is performed in the following stages: (a) We first identify `low-level diagram elements` such as the lines and arcs, `corners` i.e intersecting points of various low-level diagram elements, `objects` (e.g. block in Figure 1) and `text elements`, i.e. labels such as $\vec{F_1}, \vec{F_2}, \vec{F_3}$ and $\theta$ in Figure 1. (b) Then, we assemble the various low-level diagram elements (such as lines and arcs) to higher level diagram elements (such as axes, blocks, wedges, pulleys, etc.) by a set of human engineered grouping rules. (c) Then, we map the various text

elements to corresponding diagram elements detected in the previous stages. For example, the text element $\vec{F}_1$ in Figure 1 refers to the leftward arrow in the diagram. (d) In the final step, we use a set of human-engineered rules to maps the diagram to formal language. We show the various stages in the pipeline and their corresponding inputs and outputs in Figure 2. We will cover the details of the pipeline later in section 4. For each stage of the pipeline, we have choices for various pre-existing software (such a various line or corner detectors), rules or pre-learnt functions that we additionally wish to integrate. We also wish to minimize propagation of errors by learning the pipeline jointly. Next, we formalize the problem of learning such a pipeline.

## 3  Method

Let $x \in \mathcal{X}$ and $y \in \mathcal{Y}$ represent members of the input and output domains of a data mining task, respectively where we wish to learn a prediction function $f : \mathcal{X} \to \mathcal{Y}$ such that $\hat{y} \approx f(x)$.

**Pipeline.** We formally define a pipeline $\mathcal{P}$ as a directed acyclic graph (DAG) $G = (V, E)$ where nodes $V$ represent various computation modules in the pipeline and edges $E$ represent input/output relationships between the various modules. Given $G$, we can always derive a topological ordering of the computation modules, thus decomposing the prediction problem into $S$ stages. At each stage $s$, a predictor $f^{(s)}$ takes in as input the data instance $x$ and predictions from all previous stages $f^{(s)} : z^{(s)} \to y^{(s)}$ where $z^{(s)} = (x, \hat{y}^{(0)}, \ldots, \hat{y}^{(s-1)})$. Given a model for each stage of the pipeline, predictions are made locally and sequentially with the expressed goal of maximizing performance on all the various stages, $\hat{y} = f(x) = \{f^{(s)}(z^{(s)})\}_{s=1}^{S}$.

**Extended Pipeline.** Usually, a pipeline has a single predictor at each stage. However, system engineers are often faced with many choices for every stage of the pipeline. For example, they might have to choose between many different object detectors or many different part-of-speech taggers. It will be useful to not have to make that choice but have an ensemble of these choices. Hence, we extend our definition of the pipeline and assume that we are given multiple function approximators $\{f_i^{(s)}\}_{i=1}^{K_s}$ for the pipeline stage $s$ and we wish to use them to estimate the true underlying function $f^{(s)}$. $f_i^{(s)}$ could use a pre-existing software, encode a domain-specific rule or a pre-learnt function.

**Problem Definition:** Given the pipeline $\mathcal{P}$, multiple function approximators for each stage $\{\{f_i^{(s)}\}_{i=1}^{K_s}\}_{s=1}^{S}$ and partial supervision $\mathcal{S} = \{(x_n, \{y_n^{(t)}\}_{t \in \Omega_n})\}_{n=1}^{N}$ , we want to learn the global prediction function $f(x)$, Here, $N$ denotes the total number of data instances and $\Omega_n$ is a set of stages for which supervision is available for the $n^{th}$ data instance $x_n$. In general, at each stage, the predictor $f^{(s)}$ may output a binary prediction $y^{(s)} \in \{0, 1\}$ or a regression $y^{(s)} \in (0, 1)$. We desire a framework which can handle partial supervision, existing software and domain knowledge expressed as rules in a feasible manner. To this end, we describe `Nuts&Bolts`, a probabilistic logic framework that integrates these inputs while minimizing a global objective defined over all stages.

### 3.1  Nuts&Bolts

**Probabilistic Logic:** Probabilistic logic infers the most likely values for a set of unobserved predicates, given a set of observed ground predicate values and logic rules. Unlike classical logic, the truth values of ground predicates and rules are continuous and lie in the interval [0, 1], representing the probability that the corresponding ground predicate or rule is True. Boolean logic operators, such as AND ($\wedge$), OR ($\vee$), NOT ($\neg$) and IMPLIES ($\to$) are redefined using Lukasiewicz logic [23]: $A \wedge B = \max\{A + B - 1, 0\}$, $A \vee B = \min\{A + B, 1\}$, $\neg A = 1 - A$, and $A \to B = \min\{1 - A + B, 1\}$. Next, we define a probabilistic logic program in the most general case for learning a pipeline that integrates different kinds of function approximators, domain knowledge and partial supervision. The probabilistic logic program comprises of a model for integrating (a) multiple function approximators and (b) domain knowledge.

**A. Integrating multiple function approximators:** Based on [30], we introduce a 'trustworthiness' model to integrate multiple function approximators. Let $T_i^{(s)} \in (0, 1)$ denote how much we can `trust` the function approximator $f_i^{(s)}$. In probabilistic logic, we introduce the following rules which specify the relationship between various function approximators, the unknown true underlying function and our trusts on the various function approximators. Thus, we have:

$$f_i^{(s)}(z^{(s)}) \wedge T_i^{(s)} \to f^{(s)}(z^{(s)}), \neg f_i^{(s)}(z^{(s)}) \wedge T_i^{(s)} \to \neg f^{(s)}(z^{(s)}) \tag{1}$$

$$f_i^{(s)}(z^{(s)}) \wedge \neg T_i^{(s)} \to \neg f^{(s)}(z^{(s)}), \neg f_i^{(s)}(z^{(s)}) \wedge \neg T_i^{(s)} \to f^{(s)}(z^{(s)}) \tag{2}$$

Intuitively, the first set of rules state that if a function approximator is trustworthy, its output should match the output of the true function. The second set of rules state that if a function approximator is not trustworthy, its output should not match the output of the true function. The trust values are implicitly learnt based on the agreement between various function approximators [30].

We make an additional assumption that most of the function approximators are better than chance. With this assumption, we additionally add the following two rules: $f_i^{(s)}(z^{(s)}) \rightarrow f^{(s)}(z^{(s)}), \neg f_i^{(s)}(z^{(s)}) \rightarrow \neg f^{(s)}(z^{(s)})$. This helps alleviate the identifiability issues introduced by the above rules (eq. 1 and 2). Note that flipping the values of trusts (i.e. setting them to one minus the trust values) and the true functions leads to the rules evaluating to the same set of rules as before. In probabilistic frameworks where all the rules are weighted with a real value in $[0, 1]$, we can think of the weight of these prior belief rules as regularization weights which can be learnt from data. Note that we estimate a single trust variable for every function approximator in the pipeline stage – trust is shared across data instances. Thus, the trust variables implicitly couple various function approximators by relating them to the true underlying function, aiding semi-supervised learning.

**B. Integrating domain knowledge:** Pre-existing software or pre-learnt functions can be incorporated as function approximators in our probabilistic logic framework. Next, we will describe how we incorporate domain knowledge in the form of rules. We assume that the rules are provided to us as conditional statements (or implications) which can be read as "if `Precondition` then `Postcondition`". Note that `Precondition` and `Postcondition` can be arbitrary logical formulas (i.e. conjunctions of possibly negated predicates). In our case, the rules relate the input at a stage $z^{(s)}$ to the output $y^{(s)}$. To incorporate these rules, we introduce a function approximator for the stage $f_j^{(s)}$ and a rule $Precondition(z^{(s)}) \rightarrow f^{(s)}(z^{(s)})$. Introducing rules as function approximators allows us to combine domain knowledge expressed as rules with arbitrary function approximators using the formulation described in (A).

**C. Inference and Learning:** We use a variant of probabilistic logic, PSL [3] in our work. PSL uses soft logic as its logical component and Markov networks as its statistical model. Soft truth values of ground predicates form variables of the PSL model and the model learns weights of various rules in the logic program. Let $\mathbf{X}$ be the set of variables with known values each in the domain $[0, 1]$ and $\mathbf{Y}$ be the set of variables with unknown values in $[0, 1]$. Let $\phi = (\phi_1, \ldots, \phi_k)$ be the set of $k$ potential functions to be defined later. Given free parameters $\lambda_1, \ldots, \lambda_k$ (which correspond to the weights of various rules), we define the probability density over the set of unknown variables $\mathbf{Y}$ as:

$$f(\mathbf{Y}) = \frac{1}{Z} \exp\left( -\sum_{i=1}^{k} \lambda_i \phi_i(\mathbf{X}, \mathbf{Y}) \right)$$

$Z$ denotes the normalization constant to ensure that $f$ is a proper probability density function. Inference i.e. finding the most probable values of the unknown variables $\mathbf{Y}$ in PSL is performed by solving the convex optimization problem: $\min_{\mathbf{Y}} \sum_{i=1}^{k} \lambda_i \phi_i(\mathbf{X}, \mathbf{Y})$ solved using consensus ADMM. For learning, PSL proposes a number of approximate learning approaches such as structured perceptron, maximum-pseudolikelihood estimation and large-margin estimation. Our approach is agnostic to the choice of learning approach. In our experiments, we use maximum-pseudolikelihood estimation which maximizes the likelihood of each variable conditioned on all other variables. We refer the reader to [3] for more details. In PSL, potential functions $\phi_i$ are typically chosen to be of the form $\phi_i = (max\{0, l_i(\mathbf{X}, \mathbf{Y})\})^{p_j}$ for $p_j \in \{1, 2\}$ and $l_i$ is some linear function corresponding to a measure of the distance to satisfiability of a logic rule. According to the theory of PSL, all logic rule can be written in the form $A_1 \wedge A_2 \cdots \wedge A_u \rightarrow B_1 \vee B_2 \cdots \vee B_v$ [4]. In this case, it is easy to see that the distance to satisfiability $l_i$ can be written as $max\{0, \sum_{i=1}^{u} A_i - \sum_{i=1}^{v} B_i + 1 - u\}$ and minimizing the distance to satisfiability amounts to making the rule more satisfied. This joint modeling of the entire pipeline avoids error propagation often incurred when we sequentially use local models. The joint modeling of known and unknown variables with variable coupling through the various constraints described earlier aids semi-supervised learning as shown in our experiments.

## 4 The Diagram Parsing Pipeline

Next, we describe the various components of the diagram parsing pipeline, pointing out the pre-learnt functions, software and rules in each stage of the pipeline (see Table 1). Note that every pre-learnt function can be treated as a software. We present this difference merely due to philosophical reasons.

In the first stage, we detect low-level diagram elements (lines and arcs) using a number of pre-learnt functions and software. For corner detection, we use Harris corner detectors [14]. Then we further

Table 1: Various components of the diagram parsing pipeline. We denote the pre-learnt functions by ●, software by ● and rules by ● in each stage of the pipeline.

| | |
|---|---|
| **Line** | Apply a weak Gaussian blur on the raw image and then binarized it using a threshold selection method proposed in [26]. Then using it, apply:<br>● Boundary detection and grouping method [19]<br>● Hough transforms [9]<br>● Detect parallel curved edge segments in a canny edge map.<br>● Recursively merge proposals that exhibit a low residual when fit to a 1st or a 2nd degree polynomial.<br>● A 2 class CNN resembling VGG-16 [43] with a fourth channel (which specifies the location of the diagram element smoothed with a Gaussian kernel of width 5) appended to the standard 3 channel RGB input. |
| **Corner** | ● Harris corner detector [14] |
| **High Level** | {IF:THEN} expressions, i.e. IF 'condition' THEN 'result' rules as described below:<br>● **Arrow**: The central line (stem line) is the longest of the three lines, the two arrowhead lines are roughly of the same length, and the two angles subtended by the arrowhead lines with the arrow stem line must be roughly equal<br>● **Dotted line**: The various lines should be in a straight line, roughly the same sized lines and equi-spaced<br>● **Ground**: The solid line is in contact with a number of smaller parallel lines which subtend roughly the same angle with it, their end-point lies on the solid line and the smaller lines are on the same side with respect to the solid line<br>● **Coordinate System**: Three arrows where the arrow tails are incident on the same point. Two lines are mutually perpendicular (i.e. angle=90°) and the third roughly bisects the complementary (270°) angle<br>● **Block**: Four lines which form a rectangle<br>● **Wedge**: Three lines where each two share a distinct end-point<br>● **Pulley**: A circle with two lines tangent to it. An end-point of the two lines lies on the circle |
| **Text** | ● An off-the-shelf OCR system – `Tesseract`[3].<br>● Since many textual elements are heavily structured (these include elements in vector notation (e.g. $\vec{F}$), greek alphabets (e.g. $\theta$), physical quantities (e.g. 2 m/s)) and are usually longer than a single character, we trained a text localizer using a CNN having the same architecture as AlexNet [20]. We used the Chars74K dataset [7], a dataset obtained from vector PDFs of a number of physics textbooks and a set of synthetic renderings of structured textual elements generated by us as training data. |
| **Object** | ● Window classification [50]   ● Perceptual grouping [13, 5]   ● Cascaded ranking svm [54]<br>● Objectness [1]   ● Selective search [49]   ● Global and local search [33]   ● Edge boxes [56]<br>● A classifier with features capturing location, size, central and Hu moments, etc.<br>● A discriminatively trained part-based model [10] trained to focus on the detection of a manually selected list of objects commonly seen in physics diagrams (blocks, pulleys, etc.). |
| **Label Association** | ● **Type Matching Rules**: Type matching rules note that if the element is of type $t_1$ and the text label is of type $t_2$, then the element should be matched to the text label. Thus, the rule can be written down as $type(e, t_1) \land type(t, t_w) \to M_{to}$. We have type matching experts for the following element-object types: (a) element is an arrow and the text label is one of $F, v, a, g, x, d$ indicating physical vector quantities such as forces, velocity, acceleration and displacement, (b) element is the coordinate system and the text label is one of $x, y$ or $z$ indicating one of coordinate system axes, (c) element is a block or a wedge and the text labels it as a block' or 'wedge' (or one of their synonyms) respectively.<br>● **Proximity Rules**: The proximity rule notes that if the element and the text label are close to each other (i.e. the closest pixels of the element and the text label are closer than a threshold) then the element should be matched to the text label i.e. $proximal(t, o) \to M_{to}$<br>● **Orientation Rule**: The orientation rule notes that if the element and the text label are in the same orientation, they should be matched i.e. $orientation\_match(t, o) \to M_{to}$. The orientations are computed using the first principal component of the grey scale pixels labeled as the element/text. |
| **Formal Language** | ● Rules (one for each predicate) decide if the predicate holds for a set of diagram elements which are type consistent with the arguments of the predicate. |

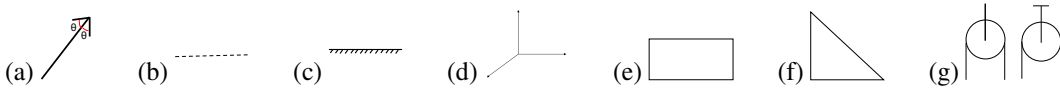

(a)     (b)     (c)     (d)     (e)     (f)     (g)

Figure 3: Some example high-level diagram elements: (a) Arrow, (b) Dotted line, (c) Ground, (d) Coordinate System, (e) Block, (f) Wedge, and (g) Pulley. We describe rules to form these elements in Table 1.

assemble these low-level diagram elements to high level elements. High level elements can be easily expressed by humans as rules given their knowledge of Physics (see Figure 3). However, it is difficult to learn the input-output mapping for high-level elements directly as this will require a very large amount of labelled data for each high-level element. We introduce a set of manually curated grouping rules for grouping low-level diagram elements to form high-level diagram elements. For example, the rule to form an arrow tests if there are three detected lines which share an end-point which can be combined to form an arrow. The three lines must also satisfy some additional conditions for the high-level element to be an arrow. The central line (stem line) is the longest of the three lines, the two arrowhead lines are roughly of the same length and the two angles subtended by the arrowhead lines with the arrow stem line must be roughly equal. This rule is incorporated as shown below:

$$C_1 = isLine(line1) \land isLine(line2) \land isLine(line3)$$
$$C_2 = length(line1) > length(line2) \land length(line1) > length(line3) \quad \textit{i.e. line1 is stem}$$
$$C_3 = roughly\_equal(angle(line1, line2), angle(line1, line3))$$
$$C_1 \land C_2 \land C_3 \to H_{line}$$

All our rules take the form of {IF:THEN} expressions, e.g. {IF condition THEN result}, where the condition tests if a set of detected low-level elements satisfy the requirements to form the high-level element. In general, we write down a rule for high-level element detection as $r_i : AND(l_{i_1}, l_{i_2}, \ldots, l_{i_\alpha}, c_{i_1}, c_{i_2}, \ldots, c_{i_\beta}) \rightarrow h_i$ s.t. the rule preconditions $P_{i_1}, P_{i_2}, \ldots, P_{i_\gamma}$ are all satisfied. Here, $l_{i_1}, l_{i_2}, \ldots, l_{i_\alpha}$ denote pre-requisite low-level elements and $c_{i_1}, c_{i_2}, \ldots, c_{i_\beta}$ denote pre-requisite corner elements required for the application of rule $r_i$ leading to the formation of high-level element $h_i$. Then, we map textual element labels with diagram elements. Let $M_{te}$ represent a variable that takes values 1 if the detected element (high-level element or object) $e$ is matched with the detected text label $t$, and 0 otherwise. Here, we have a matching constraint that $\sum_e M_{te} = 1$ which states that every text label must be matched to exactly one high-level element or object. Next, we build a set of candidate matching rules. These rules essentially capture features accounting for type, shape, orientation, distance, etc. In the final stage, we again use a set of rules to map diagrams to formal language. These rules, one for each predicate, decide if the predicate holds for a set of diagram elements which are type consistent with the arguments of the predicate. We have rules for listing objects, relative position of objects, forces acting on objects, force directions, etc.

## 5    Experiments

**Implementation:** In our implementation, we over-generate candidate low-level elements, corners, objects, text elements and plausible high-level elements from various candidate function approximators. Then, we create a set of binary variables which take value 1 if the element/corner/object/text element is correct and 0 otherwise. Then, with these variables and domain knowledge expressed as rules, we use `Nuts&Bolts` for learning the pipeline.

**Dataset:** We validated our system on a dataset of physics questions taken from three popular pre-university physics textbooks: `Resnick Halliday Walker`, `D. B. Singh` and `NCERT`. Millions of students in India study physics from these books every year and these books are available online. We manually identified chapters relevant for Newtonian physics in these textbooks and took all the exercise questions provided in these chapters. This resulted in a dataset of 4941 questions, out of which 1019 had associated diagrams. We partitioned the dataset into a random split of 4441 training (912 diagrams) and 500 test (107 diagrams) questions. We annotated ground truth logical forms for a part of the training set (1000 questions containing 207 diagrams) and the entire test set. The annotated train set questions were used along with the unannotated train set questions for training our method – thus, our method is semi-supervised. We report our results on the test set. We additionally evaluated our system on the task of answering these problems on two datasets: section 1 of three AP Physics C Mechanics tests[4] – a practice test and official tests for the years 1998 and 2012.

**Experimental Design:** We design our experimental study to evaluate the following claims:

C1: `Nuts&Bolts` outperforms prior work on the task of diagram parsing. This also leads to improvements on down stream tasks – in our case, answering Newtonian physics questions.

C2: `Nuts&Bolts` utilizes labelled data as well as unlabelled data to achieve better performance.

C3: `Nuts&Bolts` can incorporate supervision at various stages of the pipeline. It is robust to low amounts of supervision at certain stages in the pipeline.

C4: `Nuts&Bolts` jointly models the various stages of a pipeline which prevents error propagation.

**C1:**    Since there is no existing prior work which performs end-to-end parsing of diagrams to formal language, we created three baselines. The first baseline, `EB` proposes diagram elements using `EdgeBoxes` [56] and then uses rules defined in Table 1 for label associations and generation of formal language. `EdgeBoxes` was chosen as it relies less on colors and gradients observed in natural images, and because it is the only computer vision approach that performs well on all element detection stages in our experiments. The second baseline, `G-ALIGNER` proposes diagram elements using `G-ALIGNER` [40] and then uses rules. `G-ALIGNER` works by maximizing the agreement between textual and visual data via a submodular objective. Similarly, the third baseline, `DSDP-Net` uses `DSDP-Net` [17] followed by rules. `DSDP-Net` proposes diagram elements using an LSTM based scoring mechanism.

We compare the three baselines with `Nuts&Blots` in terms of the performance on predicting each type of diagram element as well as overall results in terms of the final diagram parse. We use two metrics: (a) Jaccard similarity [21] and (b) F1 score (comparing with gold annotation). Table 2

Table 2: Comparison (F1 scores and Jaccard similarity with gold annotation) for individual stages of the pipeline as well as the final parse. Certain baselines cannot be used to model some stages of the pipeline (denoted as -). We use the corresponding `Nuts&Blots` model for those stage to compute the final parse. The performance of various component function approximators is available in the supplementary.

| | F1 | | | | Jaccard Similarity | | | |
| --- | --- | --- | --- | --- | --- | --- | --- | --- |
| | EB | G-ALIGNER | DSDP-Net | **Nuts&Bolts** | EB | G-ALIGNER | DSDP-Net | **Nuts&Bolts** |
| Low-level | 0.59 | 0.78 | 0.76 | **0.94** | 0.57 | 0.80 | 0.83 | **0.87** |
| Corner | - | **0.95** | - | **0.95** | - | **0.91** | - | **0.91** |
| High-level | 0.45 | 0.71 | 0.50 | **0.90** | 0.42 | 0.74 | 0.52 | **0.82** |
| Text | 0.57 | 0.77 | 0.66 | **0.90** | 0.54 | 0.78 | 0.71 | **0.85** |
| Object | 0.33 | 0.30 | 0.52 | **0.82** | 0.29 | 0.31 | 0.47 | **0.64** |
| Label Associations | - | 0.80 | - | **0.83** | - | 0.86 | - | **0.88** |
| Parsing Performance | 0.42 | 0.65 | 0.58 | **0.74** | 0.44 | 0.68 | 0.56 | **0.78** |

reports the results. `Nuts&Blots` achieves a much superior performance to all the three baselines on both the metrics. Furthermore, we get improvements on all the stages of the pipeline. Prior computer vision techniques are tuned for natural images and hence, do not port well to diagrams which require domain knowledge. However, our carefully engineered pipeline with ensembles of element detectors and explicit domain knowledge in the form of rules can work well even in this challenging domain. **The results for problem text parsing are available in the supplementary.**

We additionally used `Nuts&Bolts` in `Parsing to Programs` [37], an existing framework proposed for "situated" question answering. The system takes in the formal representation of the problem and uses it to solve the problems using an expert system with axioms and laws of Physics written down as executable programs. We compared systems which use the `Nuts&Bolts` output against systems that use the output from various diagram parsing baselines as the formal diagram parse (all systems used `Nuts&Bolts` for text parsing). Table 3 shows the score (percentage of questions correctly answered) achieved by `Nuts&Bolts` and the various baselines on the four question answering datasets describe before. We observe that `Nuts&Bolts` achieves a better performance than all the baselines on the challenging task of answering Newtonian physics problems. **We show examples of correctly and incorrectly answered questions in the supplementary.**

| | T | P | '98 | '12 |
| --- | --- | --- | --- | --- |
| EB | 20 | 16 | 16 | 18 |
| G-ALIGNER | 23 | 19 | 16 | 21 |
| DSDP-Net | 21 | 16 | 15 | 18 |
| N&B | **32** | **24** | **24** | **26** |

Table 3: Question Answering accuracy of `Nuts&Bolts` (N&B) compared to the various baselines in the `Parsing to Programs` (P2P) framework on four datasets: problems from physics textbooks (T), AP Physics C Mechanics – Section 1 practice test (P) and official tests for 1998 and 2012.

**C2:** A key benefit of `Nuts&Blots` is its ability to incorporate unlabelled data. We investigate how changing the amount of unlabelled data in addition to the labelled data changes the performance of our diagram parser. Figure 4 plots the F1 performance of the final diagram parse as well as various stages as we vary the amount of unlabelled diagrams while keeping the labelled diagram set fixed. We can observe from the plot that adding unlabeled data substantially improves performance (from 0.58 when no unlabelled data is used to 0.74 when all unlabelled data is used). Such improvements are also observed for various stages of the pipeline to varying degrees.

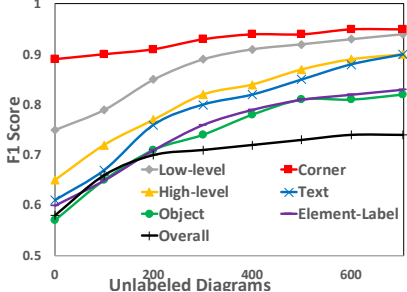

Figure 4: F1 score for diagram parsing and various pipeline stages with varying amount of unlabelled diagrams.

**C3:** `Nuts&Blots` allows for varying amount of supervision at each stage in the pipeline. Figure 5 shows the performance of individual pipeline stages as well as overall performance when we vary supervision for one stage in the pipeline (keeping supervision for all other stages same). We can observe that our model has a robust performance when supervision at an individual stage is reduced (by reducing the number of diagrams whose supervision is provided to the stage). For example, the performance merely reduces from 0.74 to 0.61 even when supervision to the text-element detection stage is reduced from 207 diagrams (the entire labelled set) to 25 diagrams.

**C4:** `Nuts&Blots` learns the entire pipeline jointly, thus preventing error propagation. To test this, we perform an ablation study using a traditional pipeline which makes sequential predictions in a stagewise manner. We consider variants of the traditional pipeline which aggregates the function

Table 4: F1 scores for identifying diagram elements, label associations and the final parse.

| | Traditional Pipeline | | | | Nuts&Bolts | | |
|---|---|---|---|---|---|---|---|
| | Best pred. | Maj. Vote | Wtd. Avg. | AdaBoost | Trust=1 | No Learn | Full |
| Low-level | 0.70 | 0.79 | 0.81 | 0.84 | 0.81 | 0.76 | **0.94** |
| Text | 0.68 | 0.77 | 0.79 | 0.80 | 0.84 | 0.80 | **0.90** |
| Object | 0.63 | 0.68 | 0.71 | 0.71 | 0.76 | 0.73 | **0.82** |
| Label Associations | 0.67 | 0.74 | 0.73 | 0.72 | 0.76 | 0.73 | **0.83** |
| Parsing Performance | 0.55 | 0.59 | 0.65 | 0.66 | 0.64 | 0.59 | **0.74** |

approximators via various combination approaches: `best predictor`, `majority vote`, `weighted average` and `AdaBoost`. We report the results in Table 4. We observe that `Nuts&Bolts` performs significantly better than each of the traditional pipeline, validating our case.

The case can be made stronger by observing Figure 5. The dotted green curve represents the performance of models that learn the combination of various function approximators separately for that stage. The solid green curve shows results for `Nuts&Bolts`. The difference between the slope of the solid green curve and the dotted green curve show that independently trained function combinators suffer much more than jointly trained function combinators with decreasing amount of available training data. Similarly, the difference between slopes of dotted blue curve and solid blue curve show how overall performance degrades when the function combinator for a particular stage is independently learned, particularly in low data scenarios.

We further investigate the importance of modelling trust in our approach. We compared `Nuts&Bolts` to a variant of `Nuts&Bolts` where we do not model trustworthiness (i.e. set all trust variables to 1). We observe a significant drop in performance which confirms the necessity of learning trust for various candidate function approximators. Then, we also show results when we do not perform learning i.e. we simply set the weights of the various rules in the PSL automatically to 1 in Table 4. We can observe that the performance of our model drops even if we do not learn rule weights of our `Nuts&Bolts` approach showing the importance of learning.

## 6    Related Work

The issue of data hungriness of end-to-end models is well known [53, 22]. Even though a number of regularization [45] and semi-supervised learning [18] techniques have been proposed, in practice data hungriness is still an issue in these models. One possible solution is incorporating domain knowledge into these models. This is also difficult and heuristic solutions such as fusing predictions from external models, feature concatenation or averaging output are popular [27]. Two promising but under-explored lines of work here are [16] and [25] who propose a teacher-student network to harness logical rules and distant supervision, respectively,

We mitigate the data hungriness issue by learning a pipeline – thereby making the learning process modular [2] and integrating domain knowledge – yet, without incurring error propagation. Our proposal is related to previous works which propose linear programming [24, 32] and graphical model [31, 15] formulations for post-hoc inference in a cascade of classifiers. A key difference in this line of work and our work is that we allow multiple function approximators in each stage in the pipeline and integration of domain knowledge in the form of rules. The former has been explored separately in various ensemble approaches [8], in error estimation from unlabeled data [28, 29, 30] and crowd-sourcing applications [46]. Our work is also related to parallel pipelines for feature

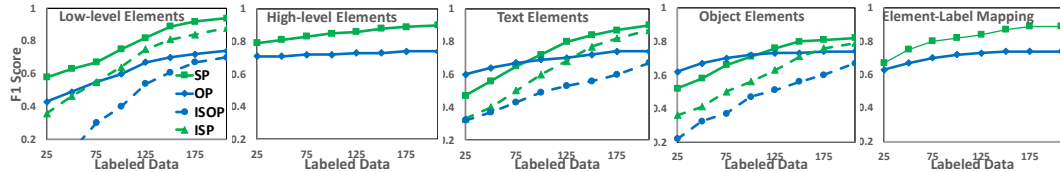

Figure 5: These plots show how the performance (F1 score for diagram parsing) varies when we vary the amount of supervision in our models for one stage in the pipeline, keeping the supervision at all other stages the same. The chart label notes the pipeline stage whose supervision is varied. For example, in the first plot we vary the amount of low-level element supervision provided to the system. The plots in solid green (SP) show the performance at the stage level, the solid blue plots (OP) show the overall performance of the system (in terms of diagram parse literals). The dotted green curve (ISP) represents the performance of models that learn the combination of various function approximators independently for that stage. The dotted blue curve (ISOP) is obtained by incorporating the independently learned aggregator into `Nuts&Bolts`.

Table 5: F1 scores for identifying diagram elements, label associations and the final parse using the bi-lattice logic formalism [42], PCFG based hierarchical grammars [55] and *Nuts&Bolts* (N&B). We have different variants for each model where we perform only inference (Inf.) where rule weights are set to 1, supervised (Sup.) and semi-supervised learning (Semi-sup.).

| | Bi-lattice logic | | PCFG | | | N&B | | |
|---|---|---|---|---|---|---|---|---|
| | Inf. | Sup. | Inf. | Supervised | Semi-sup. | Inf. | Sup. | Semi-sup. |
| Low-level | 0.65 | 0.70 | 0.62 | 0.66 | 0.77 | 0.72 | 0.75 | **0.94** |
| Corner | 0.81 | 0.85 | 0.76 | 0.79 | 0.90 | 0.86 | 0.89 | **0.95** |
| High-level | 0.57 | 0.63 | 0.54 | 0.59 | 0.66 | 0.62 | 0.65 | **0.90** |
| Text | 0.57 | 0.59 | 0.53 | 0.56 | 0.78 | 0.60 | 0.61 | **0.90** |
| Object | 0.50 | 0.53 | 0.48 | 0.51 | 0.68 | 0.53 | 0.57 | **0.82** |
| Label Associations | 0.48 | 0.54 | 0.49 | 0.55 | 0.70 | 0.52 | 0.60 | **0.83** |
| Parsing Performance | 0.49 | 0.53 | 0.47 | 0.51 | 0.64 | 0.52 | 0.58 | **0.74** |

extraction, bi-lattice logic formalism, hierarchical grammars, performance modeling and Bayesian fusion methods. These formalisms can be used in our problem setting. However, these usually do not incorporate: (a) multiple function approximators for each sub-task which is necessary when we have many weak learners but no one best model to do the sub-task, (b) existing software pieces as sub-task functions, (c) stage-wise supervision, and (d) semi-supervised learning. All these are necessary in low data scenarios when end-to-end models are infeasible. For the sake of completeness, we implemented the bi-lattice logic formalism as in [42] and a PCFG based hierarchical grammar formalism proposed in [55] where we used the best function approximator at each stage of the pipeline. PCFGs are popular in NLP and can be trained both in a supervised (via MLE) as well as semi-supervised manner (via EM). Table 5 reports F1 scores of various stages as well as the overall parsing and shows how our supervised variant outperforms the supervised learners in the bi-lattice logic formalism and PCFG. In addition, when we incorporate unlabeled data, *Nuts&Bolts* achieves a huge boost – a significant improvement over the supervised competitors and semi-supervised PCFG. Finally, our method is based on PSL [3] which allows for easy integration of first order logic rules. PSLs are a generalization of MLNs [36] which can also be used. Our approach can also be extended to other constraint driven methods such as CCMs [6] which incorporate domain knowledge in the form of constraints.

From an application perspective, while, the domain of natural images has received a lot of interest, the domain of diagram analysis is not very well studied. In particular, [12] analyzed graphs and finite automata sketches, [17] studied food-web diagrams and [40] studied geometry diagrams. As shown in our experiments, these techniques do not perform as well as `Nuts&Bolts`. This is because we leverage domain knowledge and structure of diagrams by building a bottom-up diagram parser. The idea of bottom-up analysis has been sparsely explored in images [48, 55, 44], however, without domain knowledge and use of existing softwares in the pipeline process.

# 7 Relation Extraction

We additionally also performed experiments on relation extraction (a key NLP task) comparing our approach to *Snorkel*, the recently proposed state-of-the-art approach for the task (it beat the previous best LSTM model and won the 2014 TAC-KBP challenge). We followed the same experimental protocol as in [34, 35] and compared our method to Snorkel (discriminative model – the best Snorkel model) on four datasets provided with the Snorkel release. Table 6 shows the predictive performance (in terms of Precision, Recall and F1 scores) on the relation extraction task. We use the same labeling rules as described in the Snorkel papers. *Nuts&Bolts* achieves a better performance than Snorkel on all the four datasets, and notably a new state-of-the-art. This makes a case that our framework is indeed general and widely applicable.

Table 6: Precision/Recall/F1 scores comparing Nuts&Bolts to Snorkel on four relation extraction datasets.

| | CDR | | | Spouses | | | KBP (News) | | | Genomics | | |
|---|---|---|---|---|---|---|---|---|---|---|---|---|
| | P | R | F1 | P | R | F1 | P | R | F1 | P | R | F1 |
| Snorkel | 38.8 | 54.3 | 45.3 | 48.4 | 61.6 | 54.2 | 50.5 | 29.2 | 37.0 | 83.9 | 43.4 | 57.2 |
| N&B | 41.5 | 54.5 | **47.1** | 49.3 | 61.9 | **54.9** | 51.2 | 30.3 | **38.1** | 84.5 | 43.3 | **57.3** |

# 8 Conclusion

We proposed `Nuts&Bolts`, a framework to learn pipelines with a provided hierarchy of sub-tasks. Our framework incorporates multiple function approximators for various sub-tasks, domain knowledge in the form of rules and stage-wise supervision. `Nuts&Bolts` is a philosophy of learning with modularization and can be beneficiary in limited data domains when we can learn from labelled as well as unlabelled data and when end-to-end training becomes infeasible.

**Acknowledgements**

This work is supported by the ONR grant N000141712463 and the NIH grant R01GM114311. Any opinions, findings and conclusions or recommendations expressed in this material are those of the author(s) and do not necessarily reflect the views of ONR or NIH.

## Footnotes

[1]For instance, in Figure 1, the visual concept of `floor` is represented as a solid horizontal line with many small, same sized, parallel lines with their end points on the horizontal line (see Figure 3c). This concept can be concisely expressed as a rule but is hard to learn.

[2]**Due to space constraints, we only describe the diagram parsing phase in the main paper. The methodology discussed can be readily extended to the other phases. We present the pipeline and results on text parsing and final reconciliation in the supplementary.**

[4]The other sections of the tests are subjective which we leave as future work. Details about the exam questions are available at: `https://apstudent.collegeboard.org/apcourse/ap-physics-c-mechanics`

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
