[Supplementary Material]

# Supplementary for "Learning Pipelines with Limited Data and Domain Knowledge: A Case Study in Diagram Parsing"

**Mrinmaya Sachan♣**    **Avinava Dubey♣**    **Tom Mitchell♣**    **Dan Roth♠**    **Eric P. Xing♣◇**
♣Machine Learning Department, School of Computer Science, Carnegie Mellon University
♠Department of Computer and Information Science, University of Pennsylvania
◇Petuum Inc.
`{mrinmays,akdubey,tom.mitchell,epxing}@cs.cmu.edu`
`danroth@seas.upenn.edu`

## 1   Contents

In this supplementary, we describe the text parsing component, the diagram parsing component, the reconciliation component, and the final question answering step in our model. We describe them one by one along with empirical results using our `Nuts&Bolts` approach. Finally, we also provide 10 successfully answered questions and 10 unsuccessfully answered questions to give the reader an idea of what are the sources of errors in our technique.

## 2   Text Parsing

We propose a two-stage pipeline model for question text parsing. This process is pictorially shown in Figure 1. The first stage identifies concepts in the logical language (i.e. constants, variables, functions, or predicates). In the second stage, relations are predicted with these concepts as arguments, provided some type constraints for the arguments are satisfied. For instance, the *distance* relation must take a constant which has the type *length* such as 3.00m as the second argument. Similarly, the *direction* relation must take one of the constants among $\{left, right, up, down\}$ as the second argument.

Figure 1: A pipeline for text parsing with various stages of (possibly multiple) pre-trained functions, existing software and rules.

### 2.1   Pipeline Details

Next, we describe the individual stages of the text parsing pipeline. Both stages: Concept identification and Relation identification are performed by a number of rules mentioned in Table 1. $\boldsymbol{R}_1$ represents the set of rules for concept identification using a manually curated lexicon map and a regular expression. The relation identification step again uses manually curated rules based $\boldsymbol{R}_2$ on syntax information.

In the post-processing step, we build a simple rule-based math parser to handle mathematical formulas and equations. This parser takes in a math expression such as $F = ma$ and parses it into our formal representation "equals(F, prod(m, a))". As another post procession step, we identify anaphoric and coreferential expressions in the text using Stanford CoreNLP and replace all coreferential expressions in the learnt logical formula with their corresponding antecedents.

| | | |
|---|---|---|
| $R_1$ | Lexicon Map | Indicator that the word or phrase maps to a predicate in a lexicon created by us. We derive correspondences between words/phrases and keywords and concepts in the logical language using manual annotations in the training data. For instance, our lexicon contains ("direction:left", *left*, *leftward*) including all possible realizations for the concept "direction:left". |
| | Regex for constants and explicit variables | Indicator that the word or phrase satisfies a regular expression to detect numbers or explicit variables (e.g. "3.00m", "2gm", "g m/$s^2$"). These regular expressions were built as a part of our system. |
| $R_2$ | Dependency tree distance | Shortest distance between the words of the concept nodes in the dependency tree. We use rules for distances of -3 to 3. Positive distance shows if the child word is at the right of the parent's in the sentence, and negative otherwise. |
| | Word distance | Distance between the words of the concept nodes in the sentence. We have rules for distances 0 to 3. |
| | Dependency edge | Indicator functions for outgoing edges of the parent and child for the shortest path between them. |
| | Part of speech tag | Indicator functions for the POS tags of the parent and the child |
| | Relation type | Indicator functions for unary / binary parent and child nodes. |
| | Return type | Indicator functions for the return types of the parent and the child nodes. For example, return type of *Equals* is boolean, and that of *Distance* is length. |

Table 1: The rule set for our text parsing model. We use Turboparser[1] for POS and syntactic information.

## 2.2 Results

We compared the parses induced by our models with gold parses on the test set. Table 2 reports Precision, Recall and F1 scores of the parses induced. For comparison purposes, we built a rule-based parser baseline. A similar baseline was proposed in [3] for their geometry solver. The baseline uses a set of manually designed high-precision rules. Each rule compares the dependency tree of each sentence to pre-defined templates, and if a template pattern is matched, the rule outputs the relation corresponding to that template. Our text-based parser achieved a F1 score of 0.68, a significant improvement over the rule-based parser (0.41).

| | P | R | F1 |
|---|---|---|---|
| **Rule-based** | 0.82 | 0.27 | 0.41 |
| **Nuts&Bolts** | 0.63 | 0.75 | **0.68** |

Table 2: Precision, Recall and F1 scores of parses induced by our text parser compared to a rule based parser.

We further break down this evaluation into the two components of text parsing: concept and relation identification. Table 3 shows the precision, Recall and F1 scores for concept and relation identification. Our parser achieved a high F1 score (0.91) for concept identification and a good F1 score (0.78) for relation identification.

| | P | R | F1 |
|---|---|---|---|
| **Concept** | 0.95 | 0.87 | 0.91 |
| **Relation** | 0.83 | 0.74 | 0.78 |

Table 3: Precision, Recall and F1 scores of the concept and relation identification components of our text parser.

## 3 Reconciliation of Text and Diagram Parsing

In the third phase, we reconcile text and diagram parses. This is done in many ways. For example, we incorporate question text for object detection. Often, the corresponding question texts provide important cues for detecting these visual elements. For example, the question in Figure 1 of the main document mentions the object 'trunk'. While it is unlikely that the object recognition component will correctly recognize the object 'trunk' as 'trunk' doesn't appear again in the training dataset as an object, the mention of the noun phrase 'trunk' in the question text and the context in which it appears is an important cue for identifying that this object is 'trunk'. Hence, we built a text-based object detector that uses logistic regression to classify each noun phrase in the question text as an object or not. We used a small set of manually engineered features for the prediction problem: (a) if the noun phrase is included in a list of objects manually built by us by looking at the train set, (b) if the noun phrase is an object category in ImageNet, and (c) if the noun phrase is the agent/patient (determined using the Turbo dependency parser [1]) of a small list of actions taking place in our train set (e.g. pull, run, hit ...). We included these rules in our model.

```
def vector_addition(Vectors vectors):
    result = zero_vector()
    for vector in vectors:
        result = result + vector
    return result

def angle_bw_vectors(Vector vec1, Vector vec2):
    return cos_inv(dot(vec1, vec2)/(norm(vec1)*norm(vec2)))

def project_vector(Vector vec, Direction theta):
    return (vec*cos(theta), vec*sin(theta))

def implicit_g_force(Mass m, Forces forces):
    if not forces.contains(("-mg i + 0 j")):
        forces.append(("mg i + 0 j"))

def Newton_II_law(Mass m, Forces forces, Accelerations accs):
    net_force = vector_additon(forces)
    net_acceleration = vector_addition(accs)
    return Constraint(net_force = m * net_acceleration)

def conservation_of_momentum(Mass m1, Velocity v1_initial, Mass m2, Velocity
    v2_initial, Velocity v1_final, Velocity v2_final):
    preconditions = [external_force_on_system() == None]
    if preconditions:
        return Constraint(m1*v1_initial+m2*v2_initial = m1*v1_final+m2*v2_final)
```

Figure 2: Example programs in Parsing to Programs.

Finally, we incorporate bi-modal interactions between the diagram and text parsing components by incorporating a simple rule. The rule upvotes a parse predicate if it is scored by the text as well as the diagram parser. The rule is $Pred_{diag}(p) \land Pred_{text}(p) \rightarrow Pred(p)$.

## 4 Diagram Parsing Components

**Question parsing results:** We evaluated the various question parsing components. For diagram parsing, we computed the Jaccard similarity between the diagram elements detected by our diagram parser and compared them to gold elements. We considered *Edge Boxes* [4] – since it uses edge maps to propose objects and relies less on colors and gradients observed in natural images, and our diagram parser. Table 4 reports the diagram parsing results on the test set. Our diagram parser achieved a score of 81.0 which is much better than Edge Boxes. Prior computer vision techniques are tuned for natural images and hence, do not port well to diagrams as shown in the rows colored in cyan. However, our carefully engineered pipeline with ensembles of element detectors and explicit domain knowledge in the form of rules can work well even in this challenging domain.

| Elements | E.B. | O.S. |
|----------|------|------|
| Low-level | 57.4 | 86.5 |
| 66.9, 76.2, 63.1, 74.7, 74.4 | | |
| Corner | - | 90.7 |
| High-level | 42.3 | 82.2 |
| Text | 53.6 | 85.3 |
| 76.5, 78.0 | | |
| Object | 29.1 | 63.6 |
| 43.5, 41.7, 38.5, 46.2, 40.7, 47.9, 34.6, 60.4, 61.2 | | |
| Overall | 43.8 | 81.0 |

Table 4: Jaccard similarity b/w detected diagram elements and gold elements for Edge Boxes (E.B.) and our system. We report overall results as well as results for identifying various diagram elements. For low-level, text and object element detection, we also show performance of various methods in the ensemble (cyan).

## 5 Question Answering

Subject knowledge of Newtonian physics is a crucial component in our solver. We presented the domain knowledge to the system in the form of structured programs. Some example programs are shown in Figure 2.

Some of these programs perform basic functions such as vector addition, computing angle between vectors, unit conversion, etc. Others perform more complex functions such as applying Newton's laws of motion or conservation of momentum, etc. A number of axioms denote laws of physics as a mathematical expression. For example, the Newton's second law is expressed simply as $\vec{F}_{net} = m \times \vec{a}$. Here $\vec{F}_{net}$ stands for the vector quantity representing the net force on a body. $m$ stands for the mass of the body and $\vec{a}$ stands for the acceleration of the body. These programs also define a set of preconditions which must be satisfied for it to be executable. When the preconditions are satisfied, the programs define the mathematical expression as a constraint on the output. These constraints are

then solved to obtain the answer. Parsing to Programs has a total of 237 manually curated programs. Let $\mathcal{P}$ represent this set of programs. Parsing to Programs uses this set of programs to answer the physics problems via the following deductive solver.

### 5.1 The Deductive Solver (Parsing to Programs [2])

Given access to the domain theory, we solve the physics problem by using the Parsing to Programs framework [2]. Parsing to Programs searches for program applications that can lead to the problem solution using a forward chaining search procedure exploring various possible program applications. Algorithm 1 describes the procedure.

---

**Algorithm 1:** Our Forward Chaining deductive solver

---

**Data:** Weighted set of literals $L$ representing the question and Domain knowledge $\mathcal{P}$.

1   **Do**

2    1. **Match Programs:** Match the pre-conditions of the programs against the set of literals i.e. find all programs $p \in \mathcal{P}$ s.t. the precondition $p^{pr}$ can be unified with some set of literals $L$.

3    2. **Select Program:** Sample a program (randomly uniformly) among the matching programs. Stop if no program can be applied.

4    3. **Apply Program:** Apply the chosen program by adding the result to the set of literals/constraint set.

5   **while** *#iterations* $< N_{upperbound}$;

---

The program applications are scored as a function of the scores of various literals in the program's precondition. The score of a literal is given by the confidence score from the question parser. In case it is a derived literal (derived by an earlier program execution), its score is given by the function value of the program application that derived it. Various scoring functions: minimum, arithmetic mean, geometric mean and harmonic mean were explored for all literal scores and the harmonic mean of the precondition literals performed the best, and hence is used. They further used an off-the shelf library[1] to solve the constraints introduced by the programs. Then, the following answering interface uses the search results to answer the question.

**Handling Various Question and Answer Types:** The physics examinations consist of a number of question and answer types. While a majority of questions directly ask about a particular physical quantity, there are a substantial number of questions which do not fit in this paradigm. For example, there are some *which of these are not true*, *select the odd one out*, *match the following* questions. To handle a variety of questions, Parsing to Programs has an answering interface. The interface calls the deductive solver described above and answers the question based on the type of the question or the kind of answer sought. The results on using Parsing to Programs with the Nuts&Bolts framework are already provided in the main paper.

## 6   Error Analysis

Finally, we also provide 10 successfully answered questions and 10 unsuccessfully answered questions to give the reader an idea of what are the sources of errors in our technique.

## Footnotes

[1] http://docs.sympy.org/dev/modules/solvers/solvers.html#sympy.solvers.solvers.nsolve

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

## 10 Correctly Answered Questions:

| Question | Comments |
|---|---|
| **4** You are to launch a rocket, from just above the ground, with one of the following initial velocity vectors: (1) $\vec{v}_0 = 20\hat{i} + 70\hat{j}$, (2) $\vec{v}_0 = -20\hat{i} + 70\hat{j}$, (3) $\vec{v}_0 = 20\hat{i} - 70\hat{j}$, (4) $\vec{v}_0 = -20\hat{i} - 70\hat{j}$. In your coordinate system, $x$ runs along level ground and $y$ increases upward. (a) Rank the vectors according to the launch speed of the projectile, greatest first. (b) Rank the vectors according to the time of flight of the projectile, greatest first. | (a) The speed is computed by a program for computing magnitude of the velocity. The interface ranks the results. (b) Compute the flight time using the program encoding the formula for flight time. Interface ranks the results. |
| **100** A parachutist bails out and freely falls 50 m. Then the parachute opens, and thereafter she decelerates at 2.0 m/s². She reaches the ground with a speed of 3.0 m/s. (a) How long is the parachutist in the air? (b) At what height does the fall begin? | Apply programs for both stages: free fall and deceleration,. Then solve for (a) total time and (b) total height. |
| **••32** **GO** You throw a ball toward a wall at speed 25.0 m/s and at angle $\theta_0 = 40.0°$ above the horizontal (Fig. 4-35). The wall is distance $d = 22.0$ m from the release point of the ball. (a) How far above the release point does the ball hit the wall? What are the (b) horizontal and (c) vertical components of its velocity as it hits the wall? (d) When it hits, has it passed the highest point on its trajectory? <br><br> **Fig. 4-35** Problem 32. | Apply programs that encode formulas for projectile motion to answer (a), (b) and (c). The system wrongly answers (d) due to misinterpretation of the concept "highest point". |
| **•••59** **SSM** **ILW** In Fig. 6-45, a 1.34 kg ball is connected by means of two massless strings, each of length $L = 1.70$ m, to a vertical, rotating rod. The strings are tied to the rod with separation $d = 1.70$ m and are taut. The tension in the upper string is 35 N. What are the (a) tension in the lower string, (b) magnitude of the net force $\vec{F}_{net}$ on the ball, and (c) speed of the ball? (d) What is the direction of $\vec{F}_{net}$? <br><br> Rotating rod <br><br> **Fig. 6-45** Problem 59. | Apply formula to enlist (pseudo) centripetal force. Now apply force balance and other formula theorems to solve the questions. |

**65** In Fig. 7-45, a cord runs around two massless, frictionless pulleys. A canister with mass $m = 20$ kg hangs from one pulley, and you exert a force $\vec{F}$ on the free end of the cord. (a) What must be the magnitude of $\vec{F}$ if you are to lift the canister at a constant speed? (b) To lift the canister by 2.0 cm, how far must you pull the free end of the cord? During that lift, what is the work done on the canister by (c) your force (via the cord) and (d) the gravitational force? (*Hint:* When a cord loops around a pulley as shown, it pulls on the pulley with a net force that is twice the tension in the cord.)

**Fig. 7-45** Problem 65.

All the questions can be answered by application of individual programs.

**••41** Figure 9-55 shows a two-ended "rocket" that is initially stationary on a frictionless floor, with its center at the origin of an $x$ axis. The rocket consists of a central block $C$ (of mass $M = 6.00$ kg) and blocks $L$ and $R$ (each of mass $m = 2.00$ kg) on the left and right sides. Small explosions can shoot either of the side blocks away from block $C$ and along the $x$ axis. Here is the sequence: (1) At time $t = 0$, block $L$ is shot to the left with a speed of 3.00 m/s *relative* to the velocity that the explosion gives the rest of the rocket. (2) Next, at time $t = 0.80$ s, block $R$ is shot to the right with a speed of 3.00 m/s *relative* to the velocity that block $C$ then has. At $t = 2.80$ s, what are (a) the velocity of block $C$ and (b) the position of its center?

**Fig. 9-55** Problem 41.

Assume the velocities of L and C+R after first explosion. Add constraint for velocity of L w.r.t. C+R. Assume velocities of C and R after second explosion. Add constraint for velocity of R w.r.t. C. Apply programs for conservation of momentum and solve to answer (a). Apply programs for distance moved by C to answer (b).

| | |
|---|---|
| **••57** 🌀 In Fig. 9-61, a ball of mass $m = 60$ g is shot with speed $v_i = 22$ m/s into the barrel of a spring gun of mass $M = 240$ g initially at rest on a frictionless surface. The ball sticks in the barrel at the point of maximum compression of the spring. Assume that the increase in thermal energy due to friction between the ball and the barrel is negligible. (a) What is the speed of the spring gun after the ball stops in the barrel? (b) What fraction of the initial kinetic energy of the ball is stored in the spring?<br><br>Fig. 9-61    Problem 57. | (a) Apply conservation of momentum to compute speed of the spring gun, (b) Apply conservation of energy to compute energy stored in the spring as a fraction of initial kinetic energy. |
| **53** Figure 7-41 shows a cold package of hot dogs sliding rightward across a frictionless floor through a distance $d = 20.0$ cm while three forces act on the package. Two of them are horizontal and have the magnitudes $F_1 = 5.00$ N and $F_2 = 1.00$ N; the third is angled down by $\theta = 60.0°$ and has the magnitude $F_3 = 4.00$ N. (a) For the 20.0 cm displacement, what is the *net* work done on the package by the three applied forces, the gravitational force on the package, and the normal force on the package? (b) If the package has a mass of 2.0 kg and an initial kinetic energy of 0, what is its speed at the end of the displacement?<br><br>Fig. 7-41    Problem 53. | (a) Apply programs of force addition to compute net force on the hot dogs. Then, use program to compute work, (b) Use the program that work done by forces is the change in kinetic energy. |
| **••64** 🌀 A steel ball of mass 0.500 kg is fastened to a cord that is 70.0 cm long and fixed at the far end. The ball is then released when the cord is horizontal (Fig. 9-65). At the bottom of its path, the ball strikes a 2.50 kg steel block initially at rest on a frictionless surface. The collision is elastic. Find (a) the speed of the ball and (b) the speed of the block, both just after the collision.<br><br>Fig. 9-65    Problem 64. | Apply laws of motion, conservation of momentum and energy to solve (a) and (b). |

| | |
|---|---|
| **••44** ⬤ In Fig. 9-57, a stationary block explodes into two pieces $L$ and $R$ that slide across a frictionless floor and then into regions with friction, where they stop. Piece $L$, with a mass of 2.0 kg, encounters a coefficient of kinetic friction $\mu_L$ = 0.40 and slides to a stop in distance $d_L$ = 0.15 m. Piece $R$ encounters a coefficient of kinetic friction $\mu_R$ = 0.50 and slides to a stop in distance $d_R$ = 0.25 m. What was the mass of the block? <br><br>  <br> **Fig. 9-57** Problem 44. | Apply conservation of momentum and energy to get a set of constraints. Solve to answer the question. |

## 10 Incorrectly Answered Questions:

| | |
|---|---|
| **90** A particle starts from the origin at $t = 0$ and moves along the positive $x$ axis. A graph of the velocity of the particle as a function of the time is shown in Fig. 2-43; the $v$-axis scale is set by $v_s$ = 4.0 m/s. (a) What is the coordinate of the particle at $t = 5.0$ s? (b) What is the velocity of the particle at $t = 5.0$ s? (c) What is the acceleration of the particle at $t = 5.0$ s? (d) What is the average velocity of the particle between $t = 1.0$ s and $t = 5.0$ s? (e) What is the average acceleration of the particle between $t = 1.0$ s and $t = 5.0$ s? <br><br>  <br> **Fig. 2-43** Problem 90. | The problem cannot be solved as it requires to read the plot and also compute gradient, etc. |
| **••38** A golf ball is struck at ground level. The speed of the golf ball as a function of the time is shown in Fig. 4-36, where $t = 0$ at the instant the ball is struck. The scaling on the vertical axis is set by $v_a$ = 19 m/s and $v_b$ = 31 m/s. (a) How far does the golf ball travel horizontally before returning to ground level? (b) What is the maximum height above ground level attained by the ball? <br><br>  <br> **Fig. 4-36** Problem 38. | The problem cannot be solved as it requires to read the plot and also compute area under the curve, etc. |

| | | |
|---|---|---|
| **••16** Some insects can walk below a thin rod (such as a twig) by hanging from it. Suppose that such an insect has mass $m$ and hangs from a horizontal rod as shown in Fig. 5-35, with angle $\theta = 40°$. Its six legs are all under the same tension, and the leg sections nearest the body are horizontal. (a) What is the ratio of the tension in each tibia (forepart of a leg) to the insect's weight? (b) If the insect straightens out its legs somewhat, does the tension in each tibia increase, decrease, or stay the same? |   Fig. 5-35   Problem 16. | The problem cannot be solved as it requires (a) understanding there are six such implicit legs of the insect and (b) what does "straightening" of the leg mean. |
| **•••66** Figure 5-57 shows a section of a cable-car system. The maximum permissible mass of each car with occupants is 2800 kg. The cars, riding on a support cable, are pulled by a second cable attached to the support tower on each car. Assume that the cables are taut and inclined at angle $\theta = 35°$. What is the difference in tension between adjacent sections of pull cable if the cars are at the maximum permissible mass and are being accelerated up the incline at 0.81 m/s²? |  | The problem cannot be solved as it requires to understand the phrase "adjacent sections of the pulley". |
| **••49** In Fig. 6-39, a car is driven at constant speed over a circular hill and then into a circular valley with the same radius. At the top of the hill, the normal force on the driver from the car seat is 0. The driver's mass is 70.0 kg. What is the magnitude of the normal force on the driver from the seat when the car passes through the bottom of the valley?    Fig. 6-39   Problem 49. | | The problem cannot be solved as it requires understanding the phrase "circular hill" and "circular valley". |
| **67** In Fig. 6-51, a crate slides down an inclined right-angled trough. The coefficient of kinetic friction between the crate and the trough is $\mu_k$. What is the acceleration of the crate in terms of $\mu_k$, $\theta$, and $g$?    Fig. 6-51   Problem 67. | | The problem cannot be solved as it requires 3D understanding based on two views. |

| | |
|---|---|
| •••25 GO In Fig. 7-33, a 0.250 kg block of cheese lies on the floor of a 900 kg elevator cab that is being pulled upward by a cable through distance $d_1$ = 2.40 m and then through distance $d_2$ = 10.5 m. (a) Through $d_1$, if the normal force on the block from the floor has constant magnitude $F_N$ = 3.00 N, how much work is done on the cab by the force from the cable? (b) Through $d_2$, if the work done on the cab by the (constant) force from the cable is 92.61 kJ, what is the magnitude of $F_N$? <br><br> **Fig. 7-33** Problem 25. | The problem cannot be solved as the diagram parser fails to map the objects cab and cheese. |
| 67 SSM A spring with a pointer attached is hanging next to a scale marked in millimeters. Three different packages are hung from the spring, in turn, as shown in Fig. 7-46. (a) Which mark on the scale will the pointer indicate when no package is hung from the spring? (b) What is the weight $W$ of the third package? <br><br> mm 0   40   110 N    mm 0   60   240 N    mm 0   30   $W$ <br><br> **Fig. 7-46** Problem 67. | The problem cannot be solved as it requires reading the scale and recognizing that the scale reading is a function of the extension in the spring. |
| ••35 Figure 2-24 shows a red car and a green car that move toward each other. Figure 2-25 is a graph of their motion, showing the positions $x_{g0}$ = 270 m and $x_{r0}$ = −35.0 m at time $t$ = 0. The green car has a constant speed of 20.0 m/s and the red car begins from rest. What is the acceleration magnitude of the red car? <br><br> $x$ (m), $x_{g0}$, $x_{r0}$, $t$ (s), 12 <br><br> **Fig. 2-25** Problem 35. | The problem cannot be solved as it requires reading the plot and associating with color of the plot. |

**1** Figure 4-21 shows the path taken by a skunk foraging for trash food, from initial point $i$. The skunk took the same time $T$ to go from each labeled point to the next along its path. Rank points $a$, $b$, and $c$ according to the magnitude of the average velocity of the skunk to reach them from initial point $i$, greatest first.

**Fig. 4-21** Question 1.

The problem cannot be solved as it requires reasoning based on the plot. It doesn't fall in the paradigm of programmatic solving chosen by us.