[Reviews · NeurIPS 2018]

Reviewer 1



Summary: The paper addresses an important aspect of learning pipelines with limited data while incorporating domain specific knowledge relevant to a task. The main idea is the use of PSL (probabilistic soft logic) as a framework to map partial estimates from multiple feedforward algorithms, along with domain specific logical rules, to parse visual diagrams from physics texts. Specifically, the pipelines use feature extractors for lines, arcs, corners , text elements, object elements (e.g.blocks in physics diagrams). These are combined along with human specified rules for groupings, high-level elements, text/figure labeling schemes along with the inference engine to produce the parse into a formal logical language. Experiments illustrate how the learned system: 1) is superior to state of the art diagram parsing scheme, 2) can utilize labelled as well as unlabelled data to achieve improved performance, 3) can handle various degrees of supervision in different parts of the pipeline and is robust , and 4) through integrative modeling of the stages in pipeline prevents error propagation. Quality, Clarity, originality, significance of the paper: The paper is well written and has extensive references to relevant literature, adequate experimentation. The paper is a systems application paper and adequate information is presented in supplemental material to provide details of the limitation of the results at a systems level. The current work is a systems paper combining well known aspects/algorithms in the literature. Section 5 (C1) claims that there is no prior work on end to end parsing of diagrams. While I believe this may be true for physics diagrams, the paper misses citation to key papers that exploit parallel pipelines for feature extraction, bi-lattice logic formalism for reasoning and parsing, and translation to a formal representation. See: Vinay D. Shet et al: Predicate Logic Based Image Grammars for Complex Pattern Recognition. International Journal of Computer Vision 93(2): 141-161 (2011) Toufiq Parag, Claus Bahlmann, Vinay D. Shet, Maneesh Singh: A grammar for hierarchical object descriptions in logic programs. CVPR Workshops 2012: 33-38 P. Sajda et al, In a Blink of an Eye and a Switch of a Transistor: Cortically Coupled Computer Vision, Vol. 98, No. 3, March 2010 | Proceedings of the IEEE The current paper also does not refer to a class of past work involving performance modeling of vision pipelines ( including some in the document understanding area) that involve building appropriate trust models for pipelines. Proper understanding of errors in processing stages can allow appropriate fusion mechanisms to get robust outputs (see: http://haralick.org/journals/performance.shtml). While it is clear that integrative learning at a holistic systems level reduces impact of error propagation, error modeling and appropriate bayesian fusion has been done in Bayesian model based vision literature (see: Binford, T.O., Levitt, T.S., Evidential reasoning for object recognition, PAMI(25), No. 7, July 2003, pp. 837-851. ). Given that the application domain involving physics diagrams still involve human produced diagrams, I would imagine that a systematic model-based approach can produce robust interpretations. Other comments: It will be quite difficult to reproduce the results in the paper given the complexity of the system. It is recommended that the datasets, groundtruth, and executables of the system be made available for research purposes. Comments Based on Author's rebuttal: The authors have put in substantial effort to address my concerns in my first review. Unfortunately this paper is a complex system and a lot of implementation detail for the alternatives is omitted in its present form. It is impossible to judge given the one page rebuttal why the alternative frameworks (e.g. bi-lattice reasoning) perform worse. I would want to know more details: e.g. Did they implement Shet et al paper exactly as specified? what types of rules did they use? How did they learn the rule weights? etc. The authors state that there is no prior work that uses software modules, parallel estimators, and fusion. I refer the authors to a few patents and publications in systems engineering for vision in the early 2000's address this: Methodology for model based systems engineering / performance modeling: US Patents: 7,899,209 -Statistical modeling and performance characterization of a real-time dual camera surveillance system (also appeared in: Greiffenhagen et al, CVPR 2000, CVPR 2001, Proc. of IEEE 2001) 7,428,337 - Automatic design of morphological algorithms for machine vision , by Gao et al, 2002 - addresses learning of nonlinear operator chains for a given classification task While these references are not central to the present paper, they may be useful for the authors. Overall recommendation: I don't have a problem with acceptance of the paper based on what the authors have demonstrated. While the authors definitely made attempts to address my review, comparison of alternative systems approaches are often challenging and an open question is to ask how one can make rigorous claims of superiority of one approach over another. As a systems scientist, I would not be upset with rejection of the paper. As an applications engineer, I would be ok with the acceptance of the paper. Thus, my rating for the paper does not change.

Reviewer 2



Summary This paper proposes a scheme to jointly learn components of a pipelinen in order to reap the benefis of end to end learningn (no error propagation) and learningin a pipeline( simpler tasks) simultaneously. Overall, the contribution is good, but not groundbreaking and the presentation issues bump it below the acceptance threshold for me. Strengths: The paper approaches a high level problem and takes an interesting and novel approach to doing so, integrating mutliple types of learning The paper describes a method for jointly learning a pipeline for parsing physics problems presented as images. The method is well described and the presentation is mostly clear and easy to follow. The description of the evaluation and the specific claims that the experiments are designed to assess is especially clear. Enumerating the claims and breaking analyses down by claims is easy to read and provides a good framework for discussing the paper, eg in a reading group. Weaknesses: There are a lot of presentation issues that detract from the readability of the paper. (minor) the acronym SOTA is introduced, but only used in a single paragraph; this makes for an unnecessary load on the reader to try to commi that acronym to memory, when it's not necessary to the paper; removing this would require minimal revision and improve readability, especiaclly sincne it's the frist paragraph, it is offputting to introduce non essential complexity so early line 27: is "jointly" missing prior to learning? related work as the last section makes it difficult to position the paper in context and leverage prior knowledge to understand the work, moving this to before the novel aspects would make the paper easier to read line 179: "We present this difference merely due to philosophical reasons" is dismissive of your choice of presentation; either explain why the differentiation is advantageous, even if not strictly necessary, or don't make the distinction (minor) table 1 and figure 3 visually blend together; better separation would improve readability line 207,251: missing "material" after supplementary line 283: for completeness, the name of the metric should be used instead of 'performance'. I think from the figure that it's F1score, but the text should stand alone. figure 5 x label should have units, from context I think it's samples, but sometimes similar figures use % of total and it's unclear. (minor) results tables and figure captions should contain the C# that they relate to, to connect to the structure of the text better. Notationally, the claims in the experiments section and the content at the end of page 4 should use different characters; context makes intention somewhat discernable, but unique characters will reduce ambiguity further and be useful if people aim to have a discussion about the paper PSL should be defind as an acronym in the text, not only in the citation (minor) inonsistent use of the oxoford comma, some occurences of two items joined by ",and", and other sort of random seeming uses of commas; occurencecs should each be reivewed against a single choice of style for consistent use throughout the paper the final sentence is unclear

Reviewer 3



This paper presents a case study that illustrates how to build an AI system from existing models, existing software and domain knowledge encoded as rules. This work is mostly motivated by the need to come up with techniques to learn from existing knowledge when limited amounts of data are made available. Using "nuts & bolts", the authors present a technique to build computational pipelines that join these components to solve learning tasks. They claim that learning jointly the entire pipeline addresses potential error propagation challenges that would result from concatenating components learned independently. The authors illustrate their work within a Newtonian physics setting where they learn to parse problems in this space, extracted from common textbooks. I have found this work to be quite original and interesting. The experiments are convincing and do illustrate the merit of the approach. However, the paper could greatly benefit from additional use cases beyond the Newtonian physic that is presented. In its current form, it is difficult to parse out parts of the method that are generic and parts that are specific to this use case. In general, I have found pretty much all figures very difficult to read. They are quite small. Response to rebuttal: The additional use case highlighted by the authors is very interesting and would add a lot to the paper. Indeed, the relation extraction from free text use case results that are reported are very impressive. While I understand that this submission is focusing on the nuts and bolt method, I recommend including these results in the paper. Based on this, I have reviewed my assessment for the work (from 6 to 7).